# Denitrification Characteristics of the Low-Temperature Tolerant Denitrification Strain *Achromobacter spiritinus* HS2 and Its Application

**DOI:** 10.3390/microorganisms12030451

**Published:** 2024-02-23

**Authors:** Ya-Juan Gao, Ting Zhang, Ling-Kang Hu, Shi-Yuan Liu, Chen-Chen Li, Yong-Sheng Jin, Hong-Bin Liu

**Affiliations:** 1Key Laboratory of Urban Agriculture, College of Bioscience and Resources Environment, Beijing University of Agriculture, Beijing 102206, China; gyj9901@163.com (Y.-J.G.); tingzhang@bua.edu.cn (T.Z.); hnk15585245393@163.com (L.-K.H.); shiyuanliu@bua.edu.cn (S.-Y.L.); lichenmuchen@163.com (C.-C.L.); 2Key Laboratory of Non-Point Source Pollution Control, Ministry of Agriculture and Rural Affairs/Institute of Agricultural Resources and Regional Planning, Chinese Academy of Agricultural Sciences, Beijing 100081, China

**Keywords:** *Achromobacter spiritinus*, heterotrophic nitrification, aerobic denitrification, low-temperature tolerant denitrifying bacteria, sewage treatment

## Abstract

The low-temperature environment significantly inhibits the growth and metabolism of denitrifying bacteria, leading to an excessive concentration of ammonia nitrogen and total nitrogen in sewage treatment plants during the cold season. In this study, an efficient denitrifying strain of heterotrophic nitrification–aerobic denitrification (HN–AD) bacteria named HS2 was isolated and screened from industrial sewage of a chemical factory in Inner Mongolia at 8 °C. The strain was confirmed to be *Achromobacter spiritinus*, a colorless rod-shaped bacterium. When cultured with sodium succinate as the carbon source, a carbon-to-nitrogen ratio of 20–30, a shaking rate of 150–180 r/min, and an initial pH of 6–10, the strain HS2 exhibited excellent nitrogen removal at 8 °C. Through the results of whole-genome sequencing, gene amplification, and gas product detection, the strain HS2 was determined to possess key enzyme genes in both nitrification and denitrification pathways, suggesting a HN–AD pathway of NH_4_^+^-N → NH_2_OH → NO_2_^−^N → NO → N_2_O → N_2_. At 8 °C, the strain HS2 could completely remove ammonia nitrogen from industrial sewage with an initial concentration of 127.23 mg/L. Microbial species diversity analysis of the final sewage confirmed *Achromobacter* sp. as the dominant genus, which indicated that the low-temperature denitrifying strain HS2 plays an important role in nitrogen removal in actual low-temperature sewage.

## 1. Introduction

In most northern regions of China, the outdoor temperature of sewage during the low-temperature season typically ranges from 6 to 15 degrees Celsius, and sometimes even drops below 5 degrees Celsius [1]. Most aerobic denitrifying bacteria struggle to grow under low-temperature conditions, resulting in difficulty in removing nitrogen [2,3]. Moreover, low temperatures also significantly inhibit the growth rate of bacteria and the metabolic activity of their enzymes, severely impeding the denitrification function of beneficial bacteria [4,5,6]. The cold-tolerant bacteria applied in actual sewage were shown to have significant economic and environmental benefits due to their unique cold resistance mechanisms [7]. Most studies focus on the research of HN–AD bacteria for low-concentration nitrogen sewage [8,9,10]. However, previous studies have revealed that nitrogen removal was strongly inhibited at 10 °C or less [6]. It is essential to isolate and screen HN–AD bacteria capable of effectively treating high ammonia–nitrogen sewage below 10 °C as a crucial way to solve nitrogen contamination in cold regions.

The traditional biological nitrogen removal process is distributed into aerobic and anaerobic stages. First, ammonia-oxidizing bacteria convert organic nitrogen to ammonium nitrogen under aerobic conditions; subsequently, nitrifying bacteria convert ammonium nitrogen to nitrate nitrogen. Finally, denitrifying bacteria reduce nitrate nitrogen to nitrogen under anaerobic conditions. The culture conditions of denitrifying and nitrifying bacteria are quite different, complicating the whole nitrogen removal process [11]. In addition, denitrifying bacteria are sensitive to oxygen, and nitrifying bacteria are susceptible to high ammonium, which limits their application in high-concentration ammonia sewage treatment. In 1983, researchers first proposed the concept of HN–AD, where ammonia nitrogen is converted to gaseous nitrogen by simultaneous nitrification and denitrification reactions using nutrients such as carbon sources under aerobic conditions [12]. Compared with the traditional biological method of nitrogen removal, heterotrophic nitrifying aerobic denitrifying bacteria have many advantages: the reaction occurs in the same container, and nitrification and denitrification reactions co-occur, reducing the occupied area and the applied cost. Subsequent studies have found that HN–AD microorganisms reproduce rapidly and can tolerate extreme environments such as alkaline, high temperature, low temperature, and high salinity [13]. Therefore, HN–AD bacteria are the current research hotspot in sewage nitrogen removal. However, most of the sewage nitrogen removal bacteria are not effective in actual sewage treatment, and more HN–AD microorganisms need to be screened for efficient nitrogen removal in actual sewage.

Currently, several cold-tolerant heterotrophic nitrification–aerobic denitrification bacterial genera have been reported. But most of the studied HN–AD strains currently have poor denitrification effects in actual sewage treatment. The level of cold tolerance and tolerance to high ammonia nitrogen concentrations is insufficient for practical application in sewage treatment. Therefore, it is necessary to isolate cold-tolerant HN–AD strains with stronger denitrification capabilities and study the actual application conditions of these strains for sewage denitrification under low-temperature conditions. In this work, a strain of bacteria was isolated and screened from industrial wastewater in Inner Mongolia, which could remove 100% ammonia nitrogen in the actual wastewater at 8 °C. Through the analysis of the final microbial species diversity in sewage, *Achromobacter* sp. was identified as the dominant bacterium, and the results showed that the low temperature HN–AD strain HS2 had good application value for the actual low temperature denitrification of sewage.

## 2. Materials and Methods

### 2.1. Isolation of Strain

The samples for screening cold-tolerant HN–AD bacteria were collected from a chemical sewage treatment plant in Inner Mongolia. The sewage sample was added to a sterilized enrichment medium at a seeding rate of 1% for enrichment culture. This was followed by incubation at 8 °C and 150 r/min on a constant temperature shaker for 5 days. Subsequently, 1 mL of the enriched culture was added to a new bottle of enrichment medium and incubated under the same environmental conditions on a shaker for 3 days. The composition of the enrichment medium (g/L) was as follows [14]: ammonium sulfate (4.25 g), potassium nitrate (0.72 g), brown sugar (19.02 g), trisodium citrate (8.17 g), potassium dihydrogen phosphate (3.00 g), dipotassium phosphate (8.00 g), magnesium sulfate heptahydrate (0.20 g), and trace elements solution (2.00 mL). The composition of the trace elements solution (g/L) was as follows [15]: ethylenediaminetetraacetic acid disodium salt (10.00 g), zinc sulfate heptahydrate (3.90 g), calcium chloride (7.00 g), manganese(ii) chloride tetrahydrate: (5.10 g), ferrous sulfate (5.00 g), ammonium molybdate tetrahydrate (1.10 g), copper(ii) sulfate pentahydrate (1.60 g), and cobalt chloride (1.60 g).

A quantity of 1 mL of the second-stage enriched bacterial suspension was subjected to serial dilution using the spread plate method, diluted to 10^−7^, and then spread onto isolation medium. The isolation medium was prepared by adding 20.00 g/L agar to the enriched medium. After incubation at 8 °C for 24–48 h, purified single colonies were inoculated into bromothymol blue (BTB) agar medium, inverted, and cultivated at 4 °C. We observed whether the color of these plates changed. The composition of bromothymol blue (BTB) agar medium was as follows(per liter, pH = 7.00) [14]: potassium nitrate (1.00 g), L-Asparagine (1.00 g), trisodium citrate (8.50 g), potassium dihydrogen phosphate (1.00 g), iron(III) chloride (0.05 g), calcium chloride (0.20 g), magnesium sulfate heptahydrate (1.00 g), and bromothymol blue (0.005 g).

Strains that turned from green to blue were inoculated into liquid nitrification and denitrification media. The cultures were maintained at 8 °C and 150 r/min for 5 days. Samples were collected every 24 h to measure the concentrations of TN (total nitrogen) and NH4^+^–N in each medium. The strain with the best nitrification and denitrification abilities was selected. Composition of nitrification medium (per liter, pH = 7.00) [15]: ammonium sulfate (0.944 g), sodium succinate (10.13 g), potassium dihydrogen phosphate (3.00 g), potassium hydrogen phosphate (8.00 g), magnesium sulfate heptahydrate (0.20 g), and trace elements (2.00 mL). Composition of denitrification medium (g/L) [10]: potassium nitrate (2.16 g), sodium succinate (10.13 g), potassium dihydrogen phosphate (3.00 g), potassium hydrogen phosphate (8.00 g), magnesium sulfate heptahydrate (0.20 g), and trace elements (2.00 mL).

### 2.2. Molecular Identification

A fresh culture of strain HS2 was spread onto agar plates of the isolation medium and incubated at 4 °C until colonies formed. The colony morphology was observed by a TESCAN 5136 SB scanning electron microscope (Beijing, China). Gram staining of bacterial cells was performed by a Leica DM500 optical microscope (Beijing, China). For amplification of the 16S rRNA gene sequence, a fresh culture of strain HS2 was used as a template. General primers 27F and 1492R from Sangon Biotech (Shanghai, China) were employed for polymerase chain reaction (PCR) amplification of the 16S rRNA gene in a 25 µL reaction system. The amplified product was sequenced by Sangon Biotech (Shanghai, China) and analyzed using an online BLAST program (https://www.ncbi.nlm.nih.gov/BLAST/Blast.cgi, accessed on 10 October 2021). A neighbor-joining tree of HS2 was constructed using MEGA 7.0. Subsequently, the obtained 16S rRNA gene sequence was submitted to NCBI to obtain a GENE BANK accession number.

### 2.3. Nitrification Performance under Different Temperature

To investigate the nitrogen removal performance by HS2 at low and common temperatures, the different temperatures were set at 8 °C, 16 °C, 24 °C, and 32 °C. Fresh bacteria was inoculated into the nitrification medium at a ratio of 10% (*v*/*v*), followed by constant-temperature incubation at 120 r/min. Samples of 2 mL in volumewere collected every 24 h in a clean centrifuge tube. They were then centrifuged at 12,000 r/min for 2 min at 4 °C, and the supernatant was taken for the measurement of TN, NH4^+^–N, NO_3_^−^–N, and NO_2_^−^–N concentrations.

### 2.4. Nitrification Performance under Different Culture Conditions

The factors significantly affecting the growth and nitrogen removal of strain HS2 were assessed as single-factor variables. These included carbon source, carbon/nitrogen ratio, initial pH, and shaking speed. For the optimization of nitrogen removal conditions of strain HS2 at 8 °C, a 48 h fresh bacteria culture of strain HS2 was used for inoculation. The 10% (*v*/*v*) seeding volume was added into the optimized nitrification culture medium. Carbon sources included sodium succinate, glucose, sodium acetate, and sucrose, and their C/N ratios were 5, 10, 15, and 20, respectively. Shaking speeds were set at 100, 130, 150, and 180 r/min. The initial pH of the medium was adjusted to 5.0, 6.0, 7.0, 8.0, 9.0, and 10.0 by using 1 mol/L HCl and NaOH. Samples of 2 mL in volume were collected every 24 h in a clean centrifuge. They were then centrifuged at 12,000 r/min for 2 min at 4 °C, and the supernatant was taken for the measurement of TN, NH_4_^+^–N, NO_3_^−^–N, and NO_2_^−^–N concentrations.

### 2.5. Nitrogen Balance Analysis

To analyze the nitrogen balance during the HN–AD process of strain HS2, inoculating with a 10% (*v*/*v*) seeding was added into the optimized denitrification culture medium. This was followed by incubation at 8 °C and 150 r/min on a constant temperature shaker. Sodium succinate was used as the carbon source, and the cultures were grown for 120 h. The total nitrogen at the beginning of the reaction was subtracted from the total nitrogen remaining after the reaction ended to calculate the total nitrogen removal rate of strain HS2 through denitrification. During the utilization of ammonia nitrogen by strain HS2, some nitrogen entered the cells through assimilation. After sonication, the cells of the bacteria were disrupted to release intracellular nitrogen [16,17]. The samples were crushed by an ultrasonic cell disruptor (JY92-ШD, SCIENTZ, Ningbo, China) to release the nitrogen inside the cells. The samples were then filtered through a 0.22 μm cellulose acetate membrane to obtain the filtrate used for measuring the remaining TDN, NO_3_^−^–N, NH_4_^+^–N, and NO_2_^−^–N. The calculation methods for various nitrogen concentrations are as follows:Organic nitrogen = Total Dissolved Nitrogen − Nitrate Nitrogen − Nitrite Nitrogen − Ammonia Nitrogen
Nitrogen within cells = Total Nitrogen − Total Dissolved Nitrogen
Nitrogen loss = (Initial Total Nitrogen − Remaining Nitrate Nitrogen − Remaining Nitrite Nitrogen − Remaining Ammonia Nitrogen − Remaining Organic Nitrogen − Remaining Nitrogen within Cells)/Initial Total Nitrogen × 100%.

### 2.6. The Nitrogen Removal Pathway of Strain HS2

The newly cultivated strain HS2 was used to extract genomic DNA by the TIANGEN kit method (Beijing, China). The DNA was then assessed for quality and concentration using a NanoDrop One ultramicro UV-visible spectrophotometer (nano-300, Beijing, China). The DNA samples were distributed into two parts, one for amplifying critical genes related to nitrification and denitrification, while another was used for whole-genome sequencing. Table 1 contains the primers designed for critical enzyme genes.

A volume of 1200 mL of nitrification medium was prepared in a 2 L conical flask. A 10% (*v*/*v*) inoculum of the activated strain HS2 culture was added into the nitrification medium. Pure oxygen (99.99%) was introduced into the flask and allowed to flow for approximately 5 min. The rubber tube on one side of the gas inlet was clamped, and on the other side, a gas sampling bag was attached. The flask was then placed in a shaking incubator at 8 °C and 150 r/min for cultivation. The valve on the gas sampling bag was closed after the denitrification process. The gas was qualitatively analyzed by gas chromatography–mass spectrometry(GC–MS, Agilent 7890A-5975C, Santa Clara, CA, USA).

### 2.7. The Strain HS2 Applied in Sewage

The strain HS2 was cultured in a nitrification medium and then inoculated into actual sewage via a 10% (*v*/*v*) inoculum during the logarithmic growth phase. The experiment included four groups: control group (CK), carbon source addition (carbon/TN ratio of 15), inoculation of strain HS2, and simultaneous addition of strain HS2 and carbon source. These groups were cultured under constant conditions of 8 °C and 150 r/min for aerobic cultivation. Each group was replicated three times. Samples of 2 mL in volume were collected every 24 h to measure the concentrations of total nitrogen and ammonia nitrogen.

The following steps were completed in order to further investigate the changes in the microbial community after the addition of strain HS2 to sewage. The experimental group was the SS + HS2 group, which was collected after 120 h and the microbial diversity was analyzed; the control group was the SS group. Those groups were replicated three times. The extraction and sequencing of DNA sequences were conducted by Majorbio (Shanghai, China) by the Illumina platform.

### 2.8. Analytical Analysis

The concentrations of TN, NH_4_^+^–N, NO_3_^−^–N, and NO_2_^−^–N were measured using standard methods [18]. The pH was measured using a pH meter. After appropriate dilution, the growth of the strain was measured at a wavelength of 600 nm (OD600). Data analysis and plotting were performed using Origin Pro 2021 software (Origin Lab Corp., Northampton, MA, USA).

## 3. Results

### 3.1. Isolation and Identification of Cold-Tolerant Strain

Six bacteria strains were isolated from industrial sewage at 8 °C. Based on the color changes on BTB agar medium and the determination of nitrification ability, five strains were preliminarily selected as HN–AD bacteria. Among them, the strain HS2 exhibited better abilities in heterotrophic nitrification and aerobic denitrification. Therefore, strain HS2 was chosen for further research.

As shown in Figure 1, the strain HS2 exhibited colonies with a pale yellow color, smooth and moist surface, and slight elevation, with a diameter of approximately 2–3 mm on isolation medium when cultured at 4 °C. In Figure 1C, it is shown that HS2 is Gram-negative (G−). Observation under a scanning electron microscope revealed that the bacterial cells were rod-shaped, with dimensions of approximately (2–4) μm × (0.6–1.4) μm.

The 16S rRNA sequence of strain HS2 was aligned with the GenBank database, which indicated a high similarity with *Achromobacter spiritinus*. A neighbor-joining phylogenetic tree was constructed by MEGA 7.0 software, demonstrating the closest phylogenetic relationship between strain HS2 and *Achromobacter spiritinus* and confirming the classification of strain HS2 within the *Achromobacter* genus (Figure 2). The 16S rRNA sequence of strain HS2 was submitted to NCBI, resulting in the assignment of GenBank accession number ON514055.

### 3.2. Nitrification Performance of Strain HS2 under Different Temperature

The incubation temperature was set to 8 °C, 16 °C, 24 °C, and 32 °C to examine the growth of strain HS2 in the low-temperature environment, and determine the denitrification ability of the strain over 0–96 h. The results in Figure 3 showed that strain HS2 had better nitrogen removal and growth ability at 16 °C, and strain HS2 could remove 98.84% of ammonia nitrogen and 87.73% of total nitrogen within 48 h. Strain HS2 at 8 °C still had high ability to degrade ammonia nitrogen and could degrade 95.46% of ammonia nitrogen and remove 90.80% of total nitrogen within 72 h, during which less nitrate nitrogen was detected at 6.94 mg/L. Strain HS2 at 24 °C and 32 °C could remove more than 90% of ammonia nitrogen and more than 85% of total nitrogen within 48 h. During this period, 49 mg/L of nitrate nitrogen was produced, which was subsequently removed. No nitrite nitrogen production was detected at different temperature conditions. Strain HS2 had an excellent nitrogen removal ability, which could remove over 90% of 200 mg/L ammonia nitrogen within 72 h and 100% of ammonia nitrogen at 96 h in the range of 8–32 °C.

### 3.3. Nitrification Performance under Different Culture Conditions

It was found that sodium succinate serves as the optimal carbon source for strain HS2 in Figure 4A–C. When sodium succinate was used as the carbon source, the maximum removal rates of total nitrogen and ammonium nitrogen within 120 h were 95.54% and 99.11%, respectively. There was a slight accumulation of nitrate nitrogen during this period, which was subsequently removed. No accumulation of nitrite nitrogen was detected. Citrate sodium was the second best carbon source, with removal rates of 9.14% for total nitrogen and 12.89% for ammonium nitrogen at 120 h. When glucose, sodium acetate, and sucrose were used as carbon sources, total nitrogen and ammonium nitrogen removal were negligible within a short period of time. This indicated the critical importance of selecting suitable carbon sources for strain HS2 to achieve efficient denitrification at low temperatures. Previous studies on *Halomonas piezotolerans* HN2 [19] and *Pseudomonas stutzeri sdu*10 [20] have also identified sodium succinate as the optimal carbon source. This may be attributed to the fact that succinate can enter the tricarboxylic acid (TCA) cycle directly without undergoing modification during aerobic metabolism [21]. However, the reason why strain HS2 can only utilize sodium succinate at low temperatures warrants further investigation.

From Figure 4D–F, it was evident that the optimal C/N ratio for strain HS2 was 20–30, with over 95% removal rates of total nitrogen and ammonium nitrogen within 72 h, and all experimental groups showed an increasing trend in total nitrogen and ammonium nitrogen removal rates between 24–72 h. The accumulation of nitrate nitrogen was detected with ammonium nitrogen removed, followed by the nitrate nitrogen being completely removed (Appendix A). When C/N was 10, the trend line of the ammonium nitrogen removal rate became relatively stable after 72 h. This may be attributed to an insufficient carbon source in the culture medium, which hindered bacterial growth and subsequently affected nitrogen removal by the strain [22]. Similarly, it can be observed that strain HS2 ceased growth after 72 h (Figure 4F). Therefore, strain HS2 showed a better nitrogen removal effect with C/N increasing to 20–30. This may be attributed to the higher C/N ratio in the culture medium, which allows bacteria to obtain more organic matter. It generated more energy through organic matter metabolism, thereby enhancing the nitrogen removal rate of bacteria [23,24]. However, the rate of nitrogen removal decreased when the C/N ratio was 40. This may be due to the fact that more organic matter produces more energy molecularly, such as ATP, as well as NADH, resulting in the limitation of electron transport system activity and enzyme activities caused by high levels of NADH [14].

The shaking speed determines the dissolved oxygen level in the flask. The optimal speed for strain HS2 was 180 r/min. From Figure 4G–I, it could be observed that when the speed was 180 r/min, strain HS2 achieved maximum removal rates of 90.80% for total nitrogen and 95.46% for ammonium nitrogen. The trends in total nitrogen and ammonium nitrogen removal rates at different speeds were very similar. As the speed increased, strain HS2 exhibited higher removal rates for both total nitrogen and ammonium nitrogen. This may be attributed to the higher dissolved oxygen content and better growth and metabolism of the strain, leading to enhanced nitrogen removal effects [25]. With higher speeds, there is an increased level of dissolved oxygen in the culture medium, resulting in better nitrogen removal efficiency for strain HS2. It was inferred that the presence of dissolved oxygen benefits the growth of the strain. Finally, coinciding with the completion of ammonia removal was no nitrate accumulation at all conditions. The results of *Fusarium solani* RADF-77 [26] and *Halomonas salifodinae* [27] are similar to the findings of this experiment, indicating that higher dissolved oxygen levels were conducive to denitrification by strains. This suggested that increased dissolved oxygen levels appropriately in practical low-temperature sewage treatment could enhance the nitrification performance of denitrifying bacteria.

It could be observed that the optimal pH for strain HS2 was in the range of 8–9 in Figure 4J–L. When pH was 8, removal rates of 86.17% and 91.19% were achieved for total nitrogen and ammonium nitrogen within 72 h, respectively. A few accumulations of nitrate nitrogen were observed at 48 h, and then they were removed. Between 48 and 72 h, the removal rate of total nitrogen at different initial pH levels showed an upward trend. The experimental results indicate that pH in the environment has a significant impact on denitrification by strain HS2. Both total nitrogen and ammonium nitrogen were hard to remove when the initial pH was at 5. Effective denitrification occurred within the pH range of 6–10, while the optimal pH for strain HS2 was between 8–9. Strain HS2 also exhibited a high removal rate of total nitrogen (>80%) when the pH reached 9–10, indicating its good alkaline resistance. The biological denitrification process is highly sensitive to pH values, and a weak alkaline environment (pH range of 7–9) is typically conducive to the growth of most denitrifying bacteria [2]. This finding is consistent with the results of experiments conducted with *Alcaligenes aquatilis* AS1 [21] and *Marinobacter* strain NNA5 [28], where denitrification rates were higher within the pH range of 6–9.

### 3.4. Nitrogen Balance Analysis

Under the conditions with sodium succinate as the carbon source and ammonia nitrogen as the nitrogen source, strain HS2 could completely transform 198.96 mg/L of ammonia nitrogen. As shown in Table 2, 58.36% of the ammonia nitrogen was converted into intracellular nitrogen, 16.17% was transformed into organic nitrogen existing in the reaction system, and 25.24% was converted into gaseous nitrogen. Relatively low concentrations of hydroxylamine were generated during nitrogen removal, with only 0.26% of the ammonia nitrogen converted to hydroxylamine.

During the whole nitrogen removal process, the utilization of ammonia nitrogen indicated that the strain HS2 could remove ammonia with a high rate and efficiency (83.60%) under low temperature conditions. Intracellular nitrogen via assimilation and gaseous product N_2_ via the HN–AD process accounted for 58.36% and 25.24% of the initial ammonium respectively, indicating that ammonia was utilized by strain HS2 for cell synthesis through assimilation and then the remained ammonia was converted into gaseous nitrogen via the HN–AD process at 8 °C. The yield of N_2_ by strain HS2 was similar to previous findings. *Alphaproteobacteria* W30 could convert 26.64% of the ammonium to N2 [29]. *Pseudomonas fluorescens* wsw-1001 could convert 32.99% of the ammonia nitrogen into gaseous nitrogen [30].

### 3.5. The Nitrogen Removal Pathway of Strain HS2

The extracted genomic DNA of the strain *Achromobacter spiritinus* HS2 exhibited excellent integrity, with a single and clear band observed. The concentration of the genomic DNA of strain HS2 is 123.6 ng/μL, with OD260/280 = 1.97 and OD260/230 = 1.53. These results indicate that the genomic DNA sample of strain HS2 has been successfully extracted and meets the subsequent requirements for library construction and sequencing.

The genome information of strain HS2 was presented as shown in Table 3, 23 genes involved in nitrogen metabolism were identified. These genes encompass processes such as denitrification, assimilatory nitrite reduction, nitrification, and amino acid metabolism. Further comparative analysis revealed one gene (AMO) participating in heterotrophic nitrification and four genes (NarI, NarH, NarG, and Nar) involved in aerobic denitrification pathways during the HN–AD process.

The generated gas during the HN–AD process of strain HS2 was N_2_ (Appendix A), which contradicts the result of the genome analysis. Some of the functionally relevant genes involved in nitrification and denitrification processes were found to be missing in the genome of strain HS2. Therefore, it was necessary to design relevant primers for the PCR amplification of critical enzyme genes involved in denitrification. Figure 5 shows that potential denitrification genes are detected by using the PCR method. It was successful in amplifying the functional genes related to denitrification in the genome of strain HS2. These include the nitrate reductase gene Nap (with a sequence length of 844 bp), the nitrite reductase gene Nir (with a length of 1000 bp), the nitric oxide reductase gene Nor (with a length of 432 bp), and the nitrous oxide reductase gene Nos (with a length of 275 bp). The sequencing of the products was performed, and the amplified sequences of NirK, NorB, and NorZ genes were compared to the NR database. They exhibited a similarity of up to 98% with known sequences. However, the amplified product of the NirK gene did not match any similar gene sequences in the NR database. The gaseous product N_2_ indicated that there is a gene that could convert nitrite nitrogen into gaseous nitrogen. It could be concluded that the strain HS2 possesses complete nitrification and denitrification capabilities by combining all the above gene amplification and whole-genome analysis results.

Additionally, six genes (moaA, moaD, moaE, mobB, moeA, and moeB) related to the synthesis of the cofactor molybdopterin guanine dinucleotide were found in the genome of strain HS2. Molybdopterin guanine dinucleotide constituted a part of the Nap large subunit, suggesting the presence of the Nap gene in strain HS2. Furthermore, the existence of two genes (Nir) encoding assimilatory nitrite reductase indicates the capability of strain HS2 to reduce nitrite to intracellular nitrogen. The analysis of the predicted coding genes in the HS2 whole genome revealed some genes associated with amino acid synthesis, such as glutamate dehydrogenase, glutamine synthetase, and glutamate synthase. This suggested that strain HS2 could convert ammonium nitrogen to intracellular nitrogen [31]. Specifically, the GlnA gene is involved in the conversion of ammonium nitrogen to L-Gln, while the GltB gene participates in the conversion of L-Gln to L-Glu [22]. Both L-Gln and L-Glu serve as essential energy sources for cellular biosynthesis and metabolism [32].

### 3.6. The Application of Strain HS2 in Sewage Treatment

The total nitrogen and ammonium nitrogen concentrations in the sewage were 362.45 mg/L and 127.23 mg/L, respectively. As shown in Figure 6, CK is the sewage, and it could be seen that total nitrogen and ammonia nitrogen did not change within 144 h, while strain HS2 demonstrated a main role of ammonium nitrogen removal efficacy at 8 °C in industrial sewage. In the experimental group in which HS2 was added to the sewage, nitrogen removal was promoted, which indicated that the addition of strain HS2 could be utilized for denitrification in actual sewage under low temperature conditions. At 120 h, the maximum removal rates for total nitrogen and ammonium nitrogen reached 11.34% and 43.58%, respectively. In the group with a carbon source and strain HS2, the maximum removal rates for total nitrogen and ammonium nitrogen were 38.78% and 100%, respectively. The final ammonium nitrogen concentration in the SS + HS2 group was 0 mg/L. Further improvement in the denitrification efficiency of strain HS2 could be achieved by introducing the optimal carbon source, as it was proven that carbon source was important for the growth of strain HS2.

To research the microbial community structure in sewage after the ammonia nitrogen was removed by strain HS2, bacterial community diversity analysis was performed on samples from the control group and the experimental group. Figure 7 showed the changes in microbial community at the genus level. The relative abundance of *Achromobacter* sp. was 0.18% in the control group and 70.41% in the experimental group, indicating that strain HS2 could survive in sewage for an extended period and dominate the community. Additionally, there were remarkable differences in the predominant functional genera between the two samples. In the control group, *Truepera* (14.67%) played a crucial role in nitrogen removal [33]. In the experiment group, the dominant genera were *Achromobacter* (70.41%) and *Acinetobacter* (22.12%). They all belong to *Proteobacteria*. *Proteobacteria* have been reported to encompass various denitrifying bacteria [34]. Some strains of *Acinetobacter* have been reported as low-temperature denitrifying bacteria [6], suggesting that strain HS2 not only grows rapidly in sewage but also promotes the growth of native low-temperature denitrifying bacteria.

## 4. Discussion

The current research on low-temperature nitrogen removal strains mainly focuses on the nitrogen removal performance of the strains, and there are serious questions about the effectiveness of the application for actual sewage. Strain HS2 can remove 100% of ammonia nitrogen in sewage at 8 °C. Besides, the removal of ammonia nitrogen by strain HS2 does not produce harmful intermediate products, such as nitrite and N_2_O. Many HN–AD microorganisms have emissions of N_2_O during denitrification, such as *Pseudomonas stutzeri* PCN-1 [35] and *Marinobacter hydrocarbonoclasticus* NY-4 [36]. N_2_O is a major component of greenhouse gases, so it is necessary to reduce the production of N_2_O during denitrification. *Marinobacter* strain NNA5 had zero N_2_O emissions in the same way as strain HS2 [28]. Nitrite is harmful to water organisms, so it is critical to isolate strains that have no nitrite accumulation in aquaculture sewage [37]. In contrast, the final product of ammonia nitrogen removal by *Halomonas salifodinae* Y5 is nitrite nitrogen [27], while strain HS2 would not influence the quality of the water during the removal of ammonia nitrogen. The reason is that *Halomonas salifodinae* Y5 lacks nitrite reductase genes, resulting in the inability to convert nitrite to N_2_O, which again supported the idea that HS2 has a nitrite reductase enzyme.

The nitrogen metabolism pathway of HN–AD microorganisms proposed by Wehrfritz in 1993 has been widely accepted [38]. The heterotrophic nitrification pathway is NH4^+^-N → NH_2_OH → NO_2_^−^N → NO_3_^−^-N, and the aerobic denitrification pathway is NO_3_^−^-N → NO_2_^−^N → N_2_. The denitrification pathway of strain HS2 is similar to most research findings. It was determined that *Pseudomonas aeruginosa* follows the denitrification pathway of NO_3_^−^-N → NO_2_^−^-N → NO → N_2_O → N_2_ [39]. *Pseudomonas bauzanensis* DN13-1 was found to have the same denitrification pathway. However, there is not just one nitration and denitration gene during denitrification. The enzyme of a strain might have multiple enzyme genes coexisting [40]. The strain HS2 possesses both Nap and NarG; similarly, Luz et al. [41] found that *Bacillus* spp. possessed both NiRS and NiRK nitrite reductase enzymes. There is only one nitrification gene that was detected in strain HS2, AMO, but nitrate was accumulated during the removal of ammonium nitrogen, which may be due to strain HS2 having other genes that could convert hydroxylamine to nitrate. There were various pathways for nitrification in HN–AD bacteria [42,43,44]. It would be worthwhile to determine the nitrification pathway of strain HS2 for further study, but one certain thing was that the strain could convert hydroxylamine to nitrate nitrogen (Appendix A). The nitrogen transformation pathway of HS2 could be investigated by using an isotope tracer assay to measure nitrogenous compounds or qPCR to measure the expression of genes related to the HN–AD process [45].

While strain HS2 exhibited excellent denitrification performance in sewage treatment the total nitrogen was not completely removed. This may be attributable to the complexity of sewage composition, which may hinder microbial osmoregulation and impede intracellular enzymatic activity, thereby exerting irreversible effects on strain growth and metabolism. In the later stages, enhancing the denitrification capacity of strain HS2 can be achieved through techniques such as the encapsulation of the microorganism in biopolymeric microcapsules along with organic substrates [46,47]. Additionally, the co-cultivation of fungi and bacteria could be employed to augment the denitrification efficiency of sewage [48]. To explore its application in sewage treatment, the aim is to provide applicable bacterial resources for sewage denitrification treatment under low-temperature conditions.

## 5. Conclusions

A strain of cold-tolerant HN–AD bacteria, named HS2, was isolated from industrial sewage in Inner Mongolia. The morphological characteristics and homology analysis of the 16S rRNA gene identified this strain as belonging to *Achromobacter spiritinus*. Strain HS2 exhibited the highest denitrification efficiency under the conditions of sodium succinate as the carbon source, a carbon-to-nitrogen ratio of 20–30, agitation at 150–180 rpm, and a pH range of 6–10.

It was inferred that the HN–AD pathway of strain HS2 proceeds through NH_4_^+^-N → NH_2_OH → NO_2_^−^N → NO → N_2_O → N_2_ by combining the results of whole-genome sequencing, gene amplification, and gas product detection. Under 8 °C conditions, strain HS2 was able to remove 38.78% of total nitrogen and 100% of ammonium nitrogen from industrial sewage with initial concentrations of 362.45 mg/L and 127.23 mg/L, respectively. The *Achromobacter* genus was identified as the dominant bacterium in the sewage.

## Figures and Tables

**Figure 1 microorganisms-12-00451-f001:**
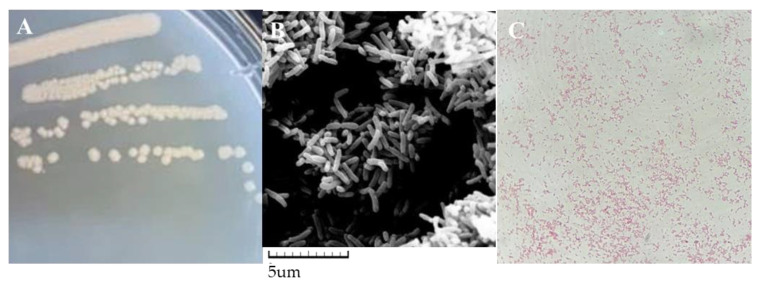
Morphological observation of strain HS2. Note: (**A**) Colony morphology image, (**B**) scanning electron microscope image, (**C**) gram staining image (10 × 100).

**Figure 2 microorganisms-12-00451-f002:**
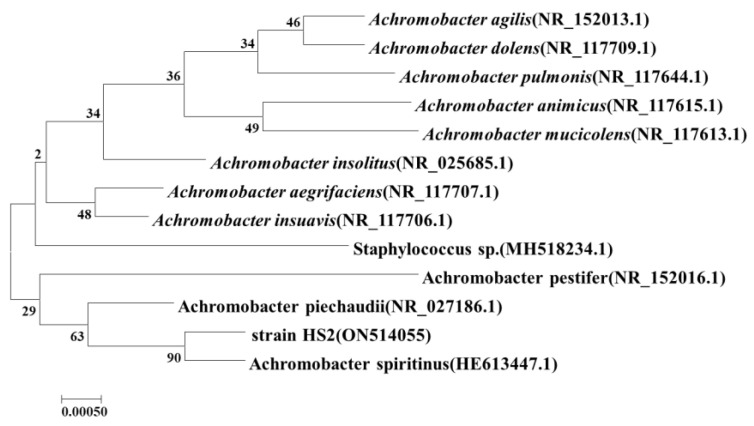
Phylogenetic tree of 16S rRNA gene sequences of strain HS2 and its closely related model strains.

**Figure 3 microorganisms-12-00451-f003:**
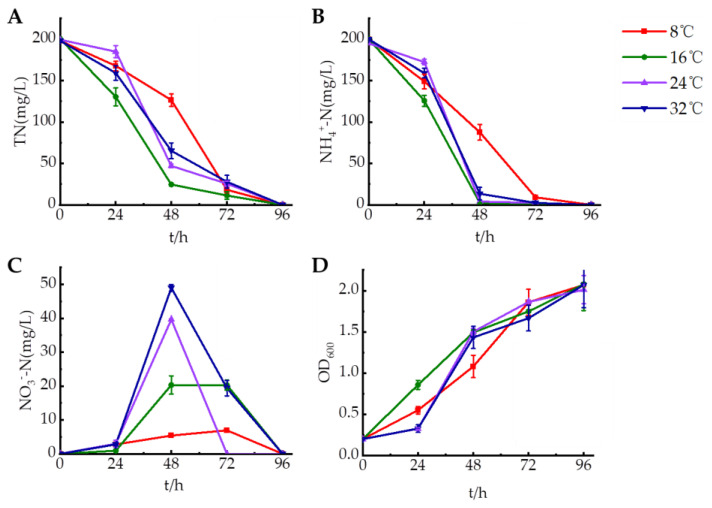
Temperature influence on the denitrification characteristics of strain HS2. (**A**): Total nitrogen, (**B**): ammonium nitrogen, (**C**): nitrate nitrogen, (**D**): OD600.

**Figure 4 microorganisms-12-00451-f004:**
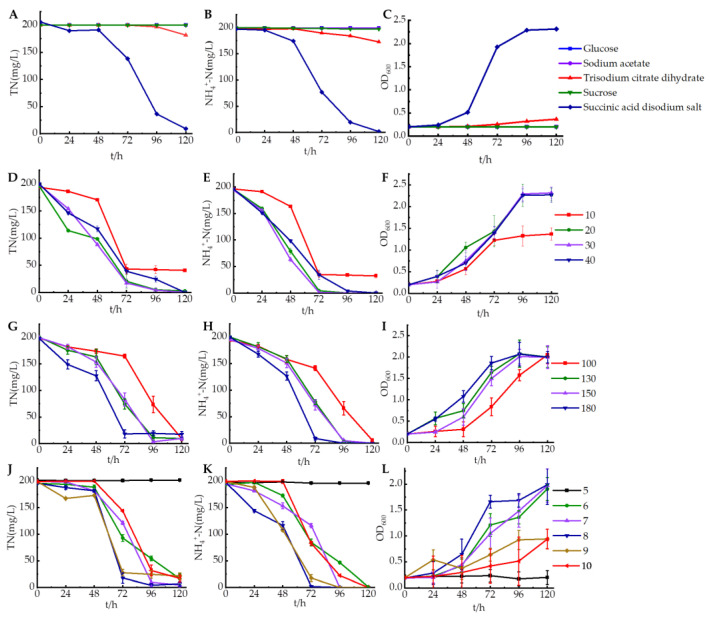
The impact of various factors on the denitrification characteristics of strain HS2. (**A**–**C**): Carbon source, (**D**–**F**): carbon-to-nitrogen ratio, (**G**–**I**): shaking speed, (**J**–**L**): pH.

**Figure 5 microorganisms-12-00451-f005:**
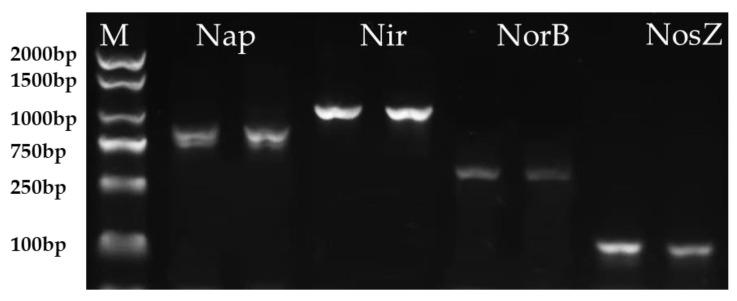
The amplification of denitrification genes of strain HS2.

**Figure 6 microorganisms-12-00451-f006:**
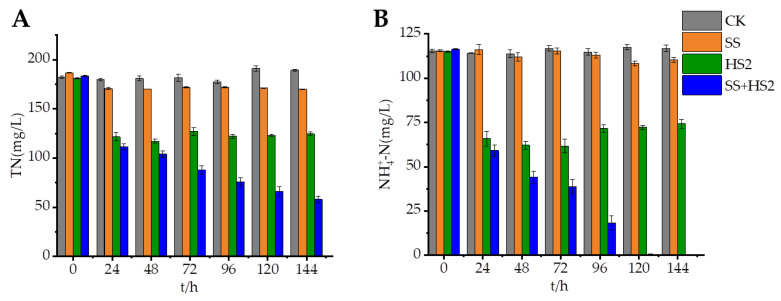
Effect of strain HS2 on the removal of total nitrogen (**A**) and ammonia nitrogen (**B**) from the chemical effluent. (SS represents the addition of sodium succinate, and SS + HS2 represents the addition of strain HS2).

**Figure 7 microorganisms-12-00451-f007:**
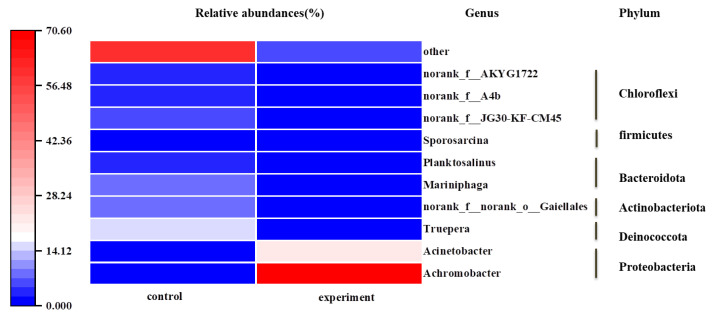
Microbial community structures after removal of ammonia nitrogen by strain HS2.

**Table 1 microorganisms-12-00451-t001:** Primers for critical enzyme genes in the denitrification process.

Gene	Primer Sequence (5′ to 3′)
16S	27F: AGAGTTTGATCMTGGCTCAG
	1492R: 5′-TACGGYTACCTTGTTACGACTT
napA	Nap1: TCTGGACCATGGGCTTCAACCA
	Nap2: ACGACGACCGGCCAGCGCAG
nirK	Nir-F: CTACTTCTCCCATCATAC
	Nir-R: CACAGGTTGTTGTTCACT
norB	Nor-F: CGNGARTTYCTSGARCARCC
	Nor-R: CRTADGCVCCRWAGAAVGC
nosZ	Nos-F: CGYTGTTCMTCGACAGCCAG
	Nos-R: CGSACCTTSTTGCCSTYGCG

**Table 2 microorganisms-12-00451-t002:** The nitrogen balance analysis for strain HS2.

Initial TN (mg/L)	Final N (mg/L)					Intracellular-N (mg/L)	N Lose (mg/L)
	NH_4_^+^–N	NH_2_OH	NO_3_^−^–N	NO_2_^−^–N	Organic N		
198.96 ± 0.86	0	0.52 ± 0.12	0	0	32.18 ± 1.71	116.12 ± 7.52	50.22 ± 6.01

**Table 3 microorganisms-12-00451-t003:** Genes involved in nitrogen metabolism in the genome of strain HS2.

Gene	Gene ID	Function
**Denitrogen**		
NarI	gene 2675	respiratory nitrate reductase subunit gamma
NarH	gene 2677	nitrate reductase subunit beta
NarG	gene 2678	nitrate reductase subunit alpha
Nar	gene 5329	nitrate reductase
MoaA	gene 2673	molybdenum cofactor biosynthesis protein MoaA
MoaD	gene 199	molybdenum cofactor biosynthesis protein
MoaE	gene 615	molybdenum cofactor biosynthesis protein
Mob B	gene 616	molybdenum cofactor biosynthesis protein B
MoeA	gene 617	molybdenumtransferase
MoeB	gene 3185	molybdopterin-synthase adenylyltransferase
**Assimilatory nitrite reduction**		
NirD	gene 776	nitrite reductase (NAD(P)H) small subunit
NirB	gene 777	nitrite reductase large subunit
**Nitrification process**		
AMO	gene 3664	ammonia monooxygenase
**Amino acid metabolism**		
GlnA	gene 1683	glutamine synthetase
GlnH	gene 4575	glutamine ABC transporter substrate-binding protein
GlnP	gene 4576	glutamine ABC transporter permease
Gud	gene 3273	NADP-specific glutamate dehydrogenase
GltA	gene 4215	glutamate synthase subunit alpha
GltB	gene 4216	glutamate synthase subunit beta

## Data Availability

All relevant data are within the paper and Appendix A.

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
