# Peer review of "Denitrification Characteristics of the Low-Temperature Tolerant Denitrification Strain Achromobacter spiritinus HS2 and Its Application"

_microorganisms, 2024, doi:10.3390/microorganisms12030451_

Round 1
Reviewer 1 Report
Comments and Suggestions for Authors
The authors are engaged in research on the process of heterotrophic nitrification and aerobic denitrification (HNAD). This process solves the shortcomings of current biological nitrogen removal systems. It is known that these systems are implemented discretely in two separate anoxic and oxic bioreactors due to the different requirements of the specifics of individual processes and the microorganisms used (slower speed of autotrophic bacteria, their sensitivity to high organic load, sensitivity of denitrifying bacteria to the presence of oxygen). Such a solution requires strict compliance with the necessary conditions, which translates into high costs. Recent research work has demonstrated the existence of some heterotrophic nitrifiers that are able to carry out nitrification using organic carbon. Several of these heterotrophic nitrifiers are able to denitrify their nitrification products, i.e. nitrates and/or nitrites, to nitrogen gas. These microorganisms can thus simultaneously breathe oxygen and nitrates and enable nitrification and denitrification processes to be carried out in one reactor under oxic conditions.
The authors of this work focused on solving the problem of the negative impact of low temperatures on the growth and metabolism of denitrifying bacteria, which leads to an excessive concentration of ammonia nitrogen and total nitrogen in wastewater treatment plants in the cold season. In this study, an efficient denitrifying strain of HAND bacteria named HS2 was isolated and screened from the industrial wastewater of a chemical factory in Inner Mongolia at 8 °C. The goal of their research was to characterize the characteristics of the low-temperature tolerant denitrifying strain Achromobacter spiritinus HS2 and its application
The results of their research show that strain HS2 showed the highest denitrification efficiency under the conditions of sodium succinate as a carbon source, carbon to nitrogen ratio of 20-30, stirring at 150-180 rpm. and pH range 6-10. At 8°C, strain HS2 was able to remove 38.78% of total nitrogen and 100% of ammonium nitrogen from industrial wastewater with initial concentrations of 362.45 mg/L TN and 127.23 mg/L NH4_N. . However, nitrogen balance experiments showed that the denitrification process by strain HS2 is primarily achieved through assimilation rather than nitrification and denitrification processes.
​
Author Response
Thank you very much for taking the time to review this manuscript. See detailed response below.
The description of these sentences is wrong. "However, nitrogen balance experiments show that the denitrification process of strain HS2 is mainly achieved through assimilation rather than nitrification and denitrification processes." Modified as follows: "During the entire denitrification process, the utilization of ammonia nitrogen shows that strain HS2 High deamination rate and efficiency (83.60%) can be achieved under low temperature conditions." The assimilated intracellular nitrogen and the gaseous product N of the HN-AD process accounted for 58.36% and 25.24% of the initial ammonium, respectively, indicating that strain HS2 through assimilation Ammonia is used for cellular synthesis and then converted to gaseous nitrogen by HN- removal. The AD process was performed at 8°C. The N production of strain HS2 was similar to previous findings.

Reviewer 2 Report
Comments and Suggestions for Authors
The manuscript is focused on the denitrification characteristics of the low-temperature tolerant denitrification strain Achromobacter spiritinus HS2 and its application.
In general, the performed work is interesting and huge. The research design is appropriate. The methods are adequately described. English is fine.
However, before the publication, the manuscript should be improved.
For this purpose, please, see the comments below:
1. The main aim of the work should be better highlighted. What is the novelty og the performed work and papers published before?
2. There is a lack of broad and in-depth analysis and discussion focused on the differences between the results presented in this manuscript and results presented in the literature.
3. What could be the practise application of the results obtained? Please, present this topic in the manuscript.
4. Figure 3 - The way of writing degrees Celsius in the legend is not consistent with the English language
Comments on the Quality of English Language
English is fine.
Author Response
Thank you very much for taking the time to review this manuscript. My response to your comments is as follows, and the revised manuscript is attached.
1.The main aim of the work should be better highlighted. What is the novelty og the performed work and papers published before?
Agree, 'This study isolated and screened a bacterium from industrial sewage in Inner Mongolia, which could remove 200mg/L of ammonia nitrogen at 8°C. The denitrification capability of strain HS2 under single-factor variables was studied, and its application in actual sewage treatment was explored, aiming to provide
applicable bacterial resources for sewage denitrification treatment under low-temperature conditions' has been modified as 'In this work, a bacteria was isolated and screened from an industrial wastewater in Inner Mongolia. Which could remove 100% ammonia nitrogen in the actual wastewater at 8℃. Through the analysis of the final microbial species diversity in sewage, Achromobacter sp was identified as the dominant bacterium, and the results showed that the low temperature HN-AD strain HS2 had good application value for the actual low temperature denitrification of sewage.'. which highlightes This emphasizes the high efficiency of HS2 in removing ammonia nitrogen from wastewater and its value for practical low-temperature sewage nitrogen removal.
Previous studies have rarely investigated the effectiveness of HN-AD bacteria for practical sewage, and this work explains that ultimately Achromobacter sp. was the most dominant genus.
2.There is a lack of broad and in-depth analysis and discussion focused on the differences between the results presented in this manuscript and results presented in the literature.
Some modifications were made to the discussion, such as explaining why HS2 did not exhibit nitrite production, unlike other strains. Additionally, unimportant examples in the discussion have been removed.
3.What could be the practise application of the results obtained? Please, present this topic in the manuscript.
The practical application aims to verify whether low-temperature-tolerant bacteria can efficiently remove ammonia nitrogen from sewage under low-temperature conditions. 'To explore its application in sewage treatment, the aim is to provide applicable bacterial resources for sewage denitrification treatment under low-temperature conditions.' has been presented in this manuscript in the discussion.
4. Figure 3 - The way of writing degrees Celsius in the legend is not consistent with the English language
The text formatting in all figures in the manuscript has been modified.

Reviewer 3 Report
Comments and Suggestions for Authors
Generally good manuscript but there are still lot of corrections and additions to be done in order to improve the quality of the presentation and make it more clear to the readers. And there are few unfinished sentences. Comments and suggestions what need to be added and corrected or explained are given in PDF document.

English language quality good but there are still few corrections that need to be done in order to be improved.
Author Response
Thank you very much for taking the time to review this manuscript. Please see the attachment, which includes the response to your comments and the revised manuscript.
